# Secondary Adrenal Insufficiency Due to Isolated ACTH Deficiency Induced by Pembrolizumab: A Report of Two Cases of Uterine Endometrial Cancer

**DOI:** 10.3390/reports6020018

**Published:** 2023-04-17

**Authors:** Ichiro Onoyama, Minoru Kawakami, Kazuhisa Hachisuga, Shoji Maenohara, Keisuke Kodama, Hiroshi Yagi, Masafumi Yasunaga, Tatsuhiro Ohgami, Kazuo Asanoma, Hideaki Yahata, Yuya Kitamura, Ryuichi Sakamoto, Daisuke Kiyozawa, Kiyoko Kato

**Affiliations:** 1Department of Obstetrics and Gynecology, Graduate School of Medical Sciences, Kyushu University, Higashi-ku, Fukuoka 812-8582, Japan; 2Department of Medicine and Bioregulatory Science, Graduate School of Medical Sciences, Kyushu University, Higashi-ku, Fukuoka 812-8582, Japan; 3Department of Anatomic Pathology, Graduate School of Medical Sciences, Kyushu University, Higashi-ku, Fukuoka 812-8582, Japan

**Keywords:** pembrolizumab, immune-related adverse event, isolated ACTH deficiency, endometrial cancer

## Abstract

Immune checkpoint inhibitors (ICIs) enhance antitumoral immune mechanisms and are used to treat various types of solid tumors including those that are microsatellite instability (MSI)-high. Uterine endometrial cancer is one of the most frequent tumor types that shows MSI-high, and, consequently, opportunities to use ICIs for endometrial cancer treatment are increasing. While using ICIs, it is important to monitor and manage various immune-related adverse events (irAEs). Here, we report two cases of secondary adrenal insufficiency during treatment of endometrial cancer with pembrolizumab. Both cases showed appetite loss and general fatigue after the 6th or 12th cycle of pembrolizumab. They were admitted to our hospital because of remarkable hyponatremia. Both cases showed no adrenocorticotropic hormone (ACTH) or cortisol response to CRH loading tests. Other pituitary hormone levels were normal, and MRI revealed no sign of hypophysitis in either patient. They were diagnosed with secondary adrenal insufficiency due to isolated ACTH deficiency and recovered soon after the administration of hydrocortisone and hydration. Thus, we should be aware of irAEs with the use of ICIs. In particular, adrenocortical insufficiency is sometimes lethal without appropriate treatment. Because the clinical symptoms are fatigue, appetite loss, and nausea, patients might be misjudged to have symptoms related to cancer. Checking serum morning cortisol before ICIs use and monitoring serum sodium levels could provide clues to diagnose secondary adrenal insufficiency.

## 1. Introduction

Immune checkpoint inhibitors (ICIs) enhance antitumoral immune mechanisms and are used to treat certain cancer types including non-small cell lung carcinoma (NSCLC), malignant melanoma, and Hodgkin lymphoma [1,2]. Pembrolizumab is an ICI, a monoclonal antibody that targets programmed cell death protein 1 (PD-1) antigen [3]. This agent is currently used to treat solid tumors including those that are MSI-high or tumor mutation burden (TMB)-high. It is well known that uterine endometrial cancer is one of the most frequent tumors that show MSI-high [4]; thus, pembrolizumab is now used in many patients with endometrial cancer and has been shown to be effective in many cases.

However, patients treated with ICIs might develop immune-related adverse events (irAEs) due to activation of the immune system. Various types of irAEs have been reported thus far, including several kinds of endocrinopathies such as hypophysitis, thyroiditis, adrenalitis, and insulitis [5]. Secondary adrenal insufficiency due to isolated adrenocorticotropic hormone (ACTH) deficiency (IAD) is one of endocrine irAEs. It is very important to diagnose IAD correctly because it is sometimes lethal without appropriate treatment. However, the incidence, predisposing factors, and prognosis of IAD remain unclear.

Here, we present two cases of IAD caused by pembrolizumab treatment of uterine endometrial cancer. Our case report will be very informative because of the paucity of IAD reports, especially in the gynecological field.

## 2. Case Presentation Section

### 2.1. Patient 1

A 67-year-old Japanese woman with uterine endometrial cancer had radical surgery (abdominal hysterectomy with bilateral adnexectomy and pelvic and para-aortic lymphadenectomy). Postoperative stage was pT1aN0M0, and pathological diagnosis was grade 3 endometrioid carcinoma (Figure 1A). Adjuvant therapy was rejected by the patient; however, at 9 months after surgery, she was found to have relapsed with multiple lung metastases. She received six cycles of paclitaxel and carboplatin chemotherapy and six cycles of doxorubicin and cisplatin chemotherapy, both of which resulted in progressive disease within 4 months after the last cycle. Then, she was started on 200 mg pembrolizumab because the primary specimen of uterine endometrial cancer was MSI-high. She received 400 mg pembrolizumab from the 5th to 12th administration, and the best overall response was 27% reduction of lung metastasis after the 12th treatment.

Three weeks after the 12th administration of pembrolizumab, she received a vaccine against COVID-19 and presented with severe general fatigue and appetite loss over 5 days. On physical examination, she was 149 cm in height and 53.6 kg in weight; her body temperature was 35.7 °C; her blood pressure was 114/90 mmHg with a regular pulse of 104/min; and her oxygen saturation was 98% on room air. Her general fatigue was severe; however, the rest of her physical examination was unremarkable. Laboratory examination revealed hyponatremia (120 mEq/L) and low serum morning cortisol (1.0 μg/dL) with a low ACTH level (5.1 pg/mL). Plasma renin activity and plasma aldosterone concentration were normal. Because she was suspected of having an endocrine irAE, such as secondary adrenal insufficiency, we performed additional examinations during hospitalization.

Magnetic resonance imaging (MRI) revealed no remarkable findings in the pituitary gland or hypothalamus. CRH and LHRH stimulation tests showed no response of ACTH and cortisol and delayed response of FSH and LH. TSH and PRL responses on TRH stimulation test were normal, and GH response was also normal on GHRP-2 test (Table 1). Finally, rapid ACTH stimulation test revealed a normal aldosterone response and a low cortisol response. The patient was diagnosed with an isolated ACTH deficiency induced by pembrolizumab from these data, although it cannot be denied that the COVID-19 vaccine was involved in her illness. She started 15 mg/day oral hydrocortisone, which was tapered to 10 mg/day after the fatigue improved and the serum sodium concentration normalized. Thereafter, she received four cycles of pembrolizumab with oral hydrocortisone. Anti-tumor effects evaluated with iRECIST were within iSD, and she is still undergoing treatment.

### 2.2. Patient 2

A 56-year-old Japanese woman was diagnosed as having uterine endometrial cancer and underwent radical surgery (abdominal hysterectomy with bilateral adnexectomy and pelvic lymphadenectomy, pT1bN0M0, grade 3 endometrioid carcinoma; Figure 1B). Paclitaxel plus carboplatin chemotherapy and doxorubicin plus cisplatin chemotherapy were administered as adjuvant chemotherapy; however, both regimens were stopped for progressive disease after three cycles and two cycles, respectively. Because the primary specimen of uterine endometrial cancer was also MSI-high in this case, she was started on 200 mg pembrolizumab. She received 400 mg pembrolizumab from the 4th to 6th administration, and the best overall response was complete remission of multiple lymph node metastasis after the 6th treatment. However, she presented with fatigue and appetite loss 6 weeks after the 6th administration.

Upon physical examination, she was 143 cm in height and 51.3 kg in weight; her body temperature was 36.2 °C; her blood pressure was 81/53 mmHg with a regular pulse of 89/min; and her oxygen saturation was 96% on room air. Laboratory examination revealed hyponatremia (135 mEq/L) and low serum morning cortisol (3.6 μg/dL) with low ACTH levels (<1.5 pg/mL). Plasma renin activity and plasma aldosterone concentration were normal. Because secondary adrenal insufficiency was also suspected in this case, we performed CRH, LHRH, and TRH stimulation tests, which revealed no response of ACTH and cortisol, a low response of FSH and LH, and a normal response of TSH and PRL (Table 2). MRI showed no remarkable findings in the pituitary gland or hypothalamus. She was diagnosed with an isolated ACTH deficiency induced by pembrolizumab and started 10 mg/day oral hydrocortisone, which improved the fatigue and normalized serum sodium concentration. After she recovered, she received four cycles of pembrolizumab administration with oral hydrocortisone. However, she could not continue the treatment because she contracted cryptococcal pneumonia. Although we had not recognized PD findings in endometrial cancer with CT, she was transferred to another hospital to manage the cryptococcal pneumonia and IAD.

## 3. Discussion

Uterine endometrial cancer is the most common gynecological malignancy. The incidence of this disease is high in countries with lower-middle incomes, in addition to industrialized countries. In 2012, the morbidity of endometrial cancer was 13,606 in Japan, which was the highest among gynecological malignancies in this country [6]. More importantly, the mortality rate of endometrial cancer has increased since the 1980s [7] despite the development of new technologies for diagnosis and treatment. One reason for the lack of improvement in therapeutic outcomes might be delays in the development of new agents for endometrial cancer. It has been more than 15 years since doxorubicin plus cisplatin chemotherapy became the first-line chemotherapy for recurrence, while paclitaxel plus carboplatin chemotherapy showed a comparable response rate [8]. In such situations, pembrolizumab, a monoclonal antibody against PD-1, has been available for advanced/recurrent solid tumors that show MSI-high or TMB-high [9,10] since 2018. Given that endometrial cancer shows MSI-high most frequently among solid tumors [4], pembrolizumab is considered a promising agent for recurrent endometrial cancer. More recently, pembrolizumab plus lenvatinib therapy [11] was approved for advanced/recurrent endometrial cancer regardless of MSI status; accordingly, the opportunities to use pembrolizumab in endometrial cancer patients are drastically increasing.

Various types of ICIs are now available, including anti-CTLA4 antibodies (ipilimumab and tremelimumab), anti-PD-1 antibodies (nivolumab and pembrolizumab), and anti-PD-L1 antibodies (atezolizumab and durvalumab). These agents are very effective for certain cancer types; however, ICIs are well known to occasionally cause autoimmune endocrine adverse effects, such as hypophysitis, thyroiditis, adrenalitis, insulitis, and parathyroiditis [12] through immune system activation. While ipilimumab is known to cause hypophysitis that accompanies pituitary enlargement, some cases of secondary adrenal insufficiency due to IAD have been reported recently. IAD is mainly caused by anti-PD-1 antibodies nivolumab and pembrolizumab and is reported most frequently in patients with lung cancer and malignant melanoma [13]. The number of IAD patients might increase as more patients receive ICI treatment. However, IAD is reported to appear in only 0.8% of patients treated with ICIs. In addition, 62.2% of patients diagnosed with IAD were male [13]. Unlike typical hypophysitis, IAD does not accompany pituitary enlargement [14]. MRI also showed no remarkable findings in the pituitary gland in our cases, although it cannot exclude that hypophysitis with long-standing adrenal insufficiency developed before and its MRI finding was recovered. MRI is recommended before ICIs use if possible.

We experienced two cases of IAD treated with pembrolizumab for recurrent endometrial cancer. Both cases need to continue oral hydrocortisone because endocrine irAEs are irreversible as reported before [15]. At our facility thus far, 5 patients have been treated with pembrolizumab and 14 patients have been treated with pembrolizumab plus lenvatinib therapy. The incidence of IAD in endometrial cancer patients is still unclear because it is only 5 years since pembrolizumab was approved for MSI-high endometrial cancer patients, and pembrolizumab plus lenvatinib therapy was approved in 2022 following the KEYNOTE-775 clinical trial [11]. Indeed, we could not find any case report regarding IAD in endometrial cancer patients. However, it cannot be denied from our data that endometrial cancer patients are more predisposed to IAD than other cancer patients when they are treated with pembrolizumab. More data are needed to determine the incidence of IAD in endometrial cancer patients.

In the case of Patient 1, she was diagnosed with IAD just after COVID-19 vaccination. COVID-19 vaccination is reported not to increase endocrine irAE incidence in those who are receiving ICIs [16]. However, in this case, sickness after COVID-19 vaccination might have manifested asymptomatic IAD that had already occurred several months before. Indeed, she showed occasional hyponatremia (minimum value 129 mEq/L) 3 months before IAD diagnosis and gradual increase in blood eosinophil percentage (maximum value 10%). Given that COVID-19 vaccination could cause adrenal crises in patients with adrenal insufficiency [17,18], we need to consider that sickness such as hyponatremia after mRNA vaccination might be a manifestation of potential IAD in patients with ICIs.

There are some predictive factors for IAD induced by ICIs. It has been reported that autoantibodies against corticotrophs, which might be produced as a reaction against ectopic expression of ACTH and Pro-opiomelanocortin (POMC) in tumors, induce IAD after ICI treatment [19]. Endometrial cancers that express ACTH or POMC are very rare [20], and immunohistochemistry revealed no ACTH expression in endometrial cancer tissues in our cases (data not shown). However, it is still possible that we might miss small populations of tumor cells that express ACTH and POMC [20]. Moreover, certain types of HLA, namely DR15, B52, and Cw12, were also reported as possible predisposing factors for pituitary irAEs [21]. Although large-scale studies are needed to confirm these data, it might become an option to examine HLA haplotypes to predict the possibility of IAD.

It was reported that hypophysitis is related to better therapeutical outcomes in malignant melanoma and NSCLC patients treated with ICIs [22], which is reasonable given that many irAEs result from autoimmune activation. However, early use of high-dose glucocorticoid for irAEs is reported to be associated with poorer survival in malignant melanoma patients [23]. In fact, pembrolizumab was very effective in both of our patients; however, Patient 2 could not continue pembrolizumab treatment because of cryptococcal pneumonia. She had poorly controlled type 2 diabetes because of poor medication adherence, which might have been involved in the onset of cryptococcal pneumonia.

## 4. Conclusions

Pembrolizumab should be monitored and managed adequately to enable long-term treatment with this approach. Based on the two cases reported here, pembrolizumab needs careful monitoring due to the potential risk of developing IAD. More clinical data, especially in endometrial cancer patients treated with ICIs, is required to enable more useful information to be given to patients and to select the best possible treatment approach.

## Figures and Tables

**Figure 1 reports-06-00018-f001:**
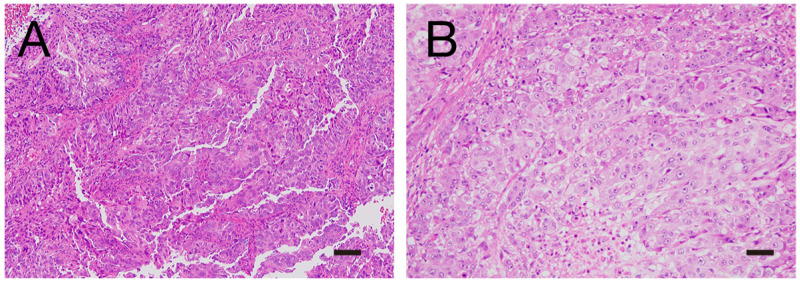
H&E staining of endometrial cancer. Representative images of Patient 1 (**A**) and Patient 2 (**B**). Bar, 50 μm.

**Table 1 reports-06-00018-t001:** Results of stimulation test of Patient 1.

Time (min)	0	15	30	60	90	120
ACTH (pg/mL)	6.3	5.9	7.1	6.9	4.5	4.4
Cortisol (μg/dL)	1.4	1.4	1.2	1.2	1	1
TSH (mIU/L)	1.8	9.6	12.7	10.3	7.7	5.8
PRL (ng/mL)	11.2	53.9	55.1	41.5	30.7	23.3
LH (mIU/L)	12.6	22.2	28.7	35.4	32	30.5
FSH (mIU/L)	33	34.4	37.8	40.4	41.9	42.1
GH (ng/mL)	1.2	10.8	12	8.2	4.9	N.E.

ACTH and cortisol levels were determined after CRH tolerance test. TSH and PRL were determined after TRH tolerance test. LH and FSH were determined after LHRH tolerance test. GH was determined by GHRP-2 test. TSH, thyroid-stimulating hormone; PRL, prolactin; LH, luteinizing hormone; FSH, follicle-stimulating hormone; GH, growth hormone; CRH, corticotropin-releasing hormone; TRH, thyrotropin-releasing hormone; LHRH, luteinizing hormone-releasing hormone; GHRP-2, growth hormone-releasing peptide 2; N.E., not examined.

**Table 2 reports-06-00018-t002:** Results of stimulation test of Patient 2.

Time (min)	0	15	30	60	90	120
ACTH (pg/mL)	<1.5	<1.5	<1.5	<1.5	<1.5	<1.5
Cortisol (μg/dL)	0.5	0.6	0.6	0.6	0.6	0.5
TSH (mIU/L)	2.9	10.6	12.7	9.6	7.2	6.0
PRL (ng/mL)	10.9	43.4	42.1	27.1	20.5	17.3
LH (mIU/L)	5.5	7.7	9.6	9.4	9.2	9.7
FSH (mIU/L)	29.7	35.9	38.8	39.4	39.5	40.3
GH (ng/mL)	2.2	40.6	30.8	19.8	15.4	N.E.

ACTH and cortisol levels were determined after CRH tolerance test. TSH and PRL were determined after TRH tolerance test. LH and FSH were determined after LHRH tolerance test. GH was determined by GHRP-2 test. TSH, thyroid-stimulating hormone; PRL, prolactin; LH, luteinizing hormone; FSH, follicle-stimulating hormone; GH, growth hormone; CRH, corticotropin-releasing hormone; TRH, thyrotropin-releasing hormone; LHRH, luteinizing hormone-releasing hormone; GHRP-2, growth hormone-releasing peptide 2; N.E., not examined.

## Data Availability

The data used in this case report are available on reasonable request from the corresponding author.

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
