# Peer review of "Secondary Adrenal Insufficiency Due to Isolated ACTH Deficiency Induced by Pembrolizumab: A Report of Two Cases of Uterine Endometrial Cancer"

_reports, 2023, doi:10.3390/reports6020018_

Round 1

Reviewer 1 Report

Congratulation on your paper.

Is a very interesting topic, but the discussion part should take in consideration the literature review in a more extensive way.

2-4 paragraphs about the results found in the literature that confirm or no the data presented should be enough.

Reviewer 2 Report

The authors report two cases of secondary adrenal insufficiency following pembrolizumab treatment in two patients with uterine endometrial cancer. While it may be valuable to raise awareness within the gynecologic oncology community about the presentation of adrenal insufficiency and the likelihood of this complication, I do not believe these cases are novel in the broader field of IRAEs where pituitary dysfunction in response to pembrolizumab has been well-characterized. For instance, A Faje et al. (European Journal of Endocrinology, 2019) reports on 22 cases of anti-PD1-induced hypophysitis in which the vast majority of patients present with fatigue and the majority with isolated ACTH deficiency without other symptoms of hypopituitarism. Faje also examines ~28 cases culled from the literature of ICI-hypophysitis secondary to PD1 inhibition and reports similar findings. As such, I do not think the cases here are particularly novel - the only difference from a sizeable number of published cases is that the patients had uterine cancer. In addition, the lack of clear imaging findings at the time of diagnosis does not exclude prior hypophysitis with long-standing AI that was diagnosed after resolution of pituitary imaging findings.

Minor comments:

Abstract:

-refers to ICI as being used to treat MSI-high cancers. While this is true, ICI are in fact used much more broadly for a variety of solid tumors. (this is also stated in the introduction)

- suggests monitoring serum sodium, which may in fact detect long-standing AI. Should AM coritsol be monitoring in these patients instead

Introduction:

- Faje et al estimate an incidence of 0.5% for hypophysitis induced by pembrolizumab - would be reasonable to include the estimate in the statement that it is "rare"

Case 1 and 2:

- As above, lack of findings on MRI do not exclude PRIOR hypophysitis. I would recommend that the authors examine the pituitary on MRI at the time of IAD diagnosis and then look at historical MRIs for the patient ot see if any pituitary shrinkage is evident as that would be suggestive of prior hypophysitis (though absence of such a finding does not exclude prior hypophysitis)

Figure 1: What tissue is stained for the patients here? Presumably this is not pituitary tissue? Why would ACTH be expected to be detectable in any other tissue? My understanding is that ACTH staining is optimized for a high expression tissue (pitutiary) and a negative result in another tissue site is not conclusive.

Discussion:

- The authors claim 2 cases out of ~19 patients treated suggests unusually high rates of IAD in patients with endometrial cancer. However, the sample size here is simply too small to speculate on this. I would encourage the authors to remove this statement and defer such conclusions until a larger sample size of treated patients can be obtained. 

Reviewer 3 Report

The case report by Ichiro Onoyama and colleagueses reported two cases of secondary adrenal insufficiency after pembrolizumab treatment for endometrial cancer. The manuscript is overall well-written and structured.

A few suggestions as follow:

1.     Line 115, “PD findings” means pathology department findings? need to be explained

2.     Line 169: “is reported most frequently in patients with lung cancer and malignant melanoma”. Literature should be cited to support this statement.

3.     Line 175-176, saying “the incidence of … seemed very high” is vague. If 5 patients were treated with pembrolizumab and 14 were treated with pembrolizumab plus lenvatinib, how many of them had IAD afterward? Only these two cases? How is the incidence that endometrial cancer patients had IAD after pembrolizumab compared with other cancer types?

4.      Line 193, “sickness after mRNA vaccination” is also too general, the authors should specify the sick symptoms such as hyponatremia because sickness after vaccination is common and maybe physiological. It couldn’t be considered a practical manifestation.

5.     Line 197, the full name of “POMC” should be mentioned somewhere in this manuscript

6.     The conclusion part needs to relate to this manuscript more. For example, based on the two cases reported here, pembrolizumab needs careful monitoring due to the potential risk of developing IAD. This is the highlight of this case report, and the authors should stress it more.

Round 2

Reviewer 2 Report

- The authors have addressed many of my concerns.

- I am still confused about the images of endometrial cancer and staining for ACTH - the relevant staining would be in pituitary and one would expect these patients to have no ACTH staining in pituitary. Of course, I recognize that tissue is not available. ACTH is not normally expressed in endometrium or, to my knowledge, endometrial cancer, so it is really meaningless to show negative ACTH staining of the endometrial cancer in these cases. (And I suspect the same result would be seen in individuals who do not have IAD.) I would recommend removing these images.
